# Early Postoperative Low Compliance to Enhanced Recovery Pathway in Rectal Cancer Patients

**DOI:** 10.3390/cancers14235736

**Published:** 2022-11-22

**Authors:** Marco Ceresoli, Corrado Pedrazzani, Luca Pellegrino, Andrea Muratore, Ferdinando Ficari, Roberto Polastri, Marco Scatizzi, Mauro Totis, Nicolò Tamini, Lorenzo Ripamonti, Marco Braga

**Affiliations:** 1Department of General and Emergency Surgery, School of Medicine and Surgery, University of Milano-Bicocca, 20900 Monza, Italy; 2Department of General Surgery, University of Verona, 37100 Verona, Italy; 3Department of General Surgery, Candiolo Cancer Institute–FPO–IRCCS, 10060 Candiolo, Italy; 4Department of General Surgery, Pinerolo Hospital, 10064 Pinerolo, Italy; 5Department of General Surgery, Careggi Hospital, University of Firenze, 50100 Firenze, Italy; 6Department of Surgery, Degli Infermi Hospital, 13900 Biella, Italy; 7Department of General Surgery, Santa Maria Annunziata ASL Toscana Centro, 50100 Firenze, Italy

**Keywords:** rectal surgery, enhanced recovery, overall morbidity, ERAS compliance, low pneumoperitoneum, TAP block

## Abstract

**Simple Summary:**

This research investigates the adherence and compliance to the ERAS pathway in patients operated for rectal cancer; the results highlights the important role of early postoperative compliance to the postoperative pathway with the development of complications.

**Abstract:**

Early postoperative low compliance to enhanced recovery protocols has been associated with morbidity following colon surgery. The purpose of this study is to evaluate the possible causes of early postoperative low compliance to the enhanced recovery pathway and its relationship with morbidity following rectal surgery for cancer. A total of 439 consecutive patients who underwent elective surgery for rectal cancer have been included in the study. Compliance to enhanced recovery protocol on postoperative day (POD) 2 was evaluated in all patients. Indicators of compliance were naso-gastric tube and urinary catheter removal, recovery of both oral feeding and mobilization, and the stopping of intravenous fluids. Low compliance on POD 2 was defined as non- adherence to two or more items. One-third of patients had low compliance on POD 2. Removal of urinary catheter, intravenous fluids stop, and mobilization were the items with lowest adherence. Advanced age, duration of surgery, open surgery and diverting stoma were predictive factors of low compliance at multivariate analysis. Overall morbidity and major complications were significantly higher (*p* < 0.001) in patients with low compliance on POD 2. At multivariate analysis, failure to remove urinary catheter on POD 2 (OR = 1.83) was significantly correlated with postoperative complications. Low compliance to enhanced recovery protocol on POD 2 was significantly associated with morbidity. Failure to remove the urinary catheter was the most predictive indicator. Advanced age, long procedure, open surgery and diverting stoma were independent predictive factors of low compliance.

## 1. Introduction

Enhanced recovery protocols have been associated with a significant improvement of outcome after major surgery for gastrointestinal cancer [1,2,3]. The elderly and patients with multiple comorbidities can be included in the enhanced recovery program, but often require a tailored protocol [4,5].

Early postoperative low compliance to an enhanced recovery protocol has been reported in about one third of patients following elective colonic resection [6,7]. Patients with early low compliance after colonic resection had significant higher morbidity and longer hospital stays [8]. Few data are currently available on both the rate and causes of early low compliance to enhanced recovery protocols following rectal surgery and its relationship with morbidity occurring afterwards.

The first experiences with enhanced recovery protocols were carried out more than 20 years ago. New items reducing perioperative stress and invasiveness of surgery have been subsequently proposed [9,10]. Promising preliminary results have been obtained with low-pressure pneumoperitoneum, multimodal analgesia including abdominal wall blocks, and inferior mesenteric artery preservation in upper rectal cancer surgery [11,12,13,14].

The purpose of this study is to assess which variables can be associated with low compliance to enhanced recovery pathways. The relationship between low compliance and overall postoperative morbidity has also been investigated.

## 2. Materials and Methods

The present study is performed in accordance with STROBE guidelines [15]. Consecutive patients who underwent elective surgery for rectal cancer in seven Italian hospitals have been included in the study. Patients with combined resections (rectal and other viscera) were excluded. All patients have been prospectively registered in the database of the PeriOperative Italian Society. Each hospital applied a comprehensive ERAS pathway according to the ERAS^®^ Society recommendation in colorectal surgery [16] and followed a pathway implementation program before starting the study [8].

In 35 consecutive patients who underwent surgery for upper rectal cancer at Monza Hospital, Monza, Italy, (Monza subgroup), three operative items have been added to the study protocol: TAP (transversus abdominis plane) block instead thoracic epidural catheter, low pneumoperitoneum (8 mmHg), and inferior mesenteric artery (IMA) sparing.

Demographics, perioperative variables, adherence to each item of the protocol, and short-term outcome parameters were prospectively collected in all patients. Indicators of postoperative compliance were naso-gastric tube and urinary catheter removal, recovery of oral feeding and mobilization out of bed, and the stopping of intravenous fluids. Removal of the naso-gastric tube was planned at the end of surgery; and patients were mobilized the day of surgery. The starting of oral feeding and removal of urinary catheter were planned on postoperative day 1. Intravenous fluid infusion was discontinued as early as possible in accordance with the recovery of oral feeding. Low compliance on postoperative day (POD) 2 was defined as non-adherence to two or more items [8].

Criteria to identify each postoperative complication were defined a priori [17] and the Clavien-Dindo classification has been used to grade their severity [18]. Complications graded as IIIb to V were considered as major. Discharge criteria and time to readiness for discharge were defined according to a previous study [19]. Any hospital readmission due to postoperative complications occurring within 30 days after discharge has been registered.

### Statistical Analysis

Continuous variables were reported as median along with the interquartile range (IQR) and compared with a Mann-Whitney’s U test, while categorical variables were reported as percentages and compared with the Chi square test. Variables predictive of complications were individuated with uni and multiple logistic regression methods. The analysis of factors associated with low compliance on POD2 was carried out in uneventful patients. Statistics were performed with SPSS 25 (IBM Corp. Released 2017, IBM SPSS Statistics for Windows, Version 25.0. IBM Corp: Armonk, NY, USA).

## 3. Results

The present analysis includes 439 consecutive cancer patients who underwent elective rectal resection. An (American Society of Anesthesiology) ASA score of 3–4 was found in 141 (32.1%) patients, neoadjuvant chemo-radiotherapy was carried out in 113 (25.7%) patients, and laparoscopic surgery was successfully performed in 373 (82.7%) patients (Table 1).

The overall adherence to preoperative and operative items was 81.3%. No patient received oral antibiotics before surgery. Mechanical bowel preparation was carried out in 172 (39.3%) patients, while an abdominal drain was placed in 345 (78.8%). The naso-gastric tube was removed at the end of surgery in 398 (90.8%) patients.

Table 2 shows that a low protocol compliance on POD 2 was found in one-third of patients. The items with the lowest adherence were removal of the urinary catheter, the stopping of intravenous fluids, and mobilization.

Table 3 shows that advanced age, long surgical procedure, open surgery, and diverting stoma were significantly associated to low compliance on POD 2, whereas operative volemia monitoring was associated with high compliance (*p* = 0.06)**.**

Table 4 reports short-term outcomes. Postoperative morbidity occurred in 149 (32.6%) patients and major complications occurred in 27 (6.2%) patients. Twenty-four (5.5%) patients underwent reoperation. Median time to readiness for discharge and length of hospital stay were 5 (4–8) and 6 (5–8) days. The readmission rate was 3.0% (13 patients).

Figure 1 shows that patients with low compliance on POD 2 had higher overall morbidity and major complications. At multivariate analysis, failure to remove the urinary catheter on POD 2 was significantly correlated with postoperative complications (Table 5).

Table 6 reports data on patients of the Monza subgroup who have a higher rate of ASA 3 compared to the overall series. The TAP block and IMA sparing technique were successfully performed in all patients, while low pneumoperitoneum failed in 5 (14.2%) patients who needed an increase up to 12 mmHg. Lymph-node collection and postoperative pain score were similar to the overall series, while early mobilization was observed in 32 (91.4%) patients. No anastomotic leak occurred.

## 4. Discussion

Low compliance to an enhanced recovery protocol was found in about one- third of patients after rectal surgery. Patients with low compliance on POD 2 had higher overall morbidity and major complications. Variables associated with early low compliance were advanced age, long procedure, open surgery, and diverting stoma. Upon multivariate analysis, failure to remove the urinary catheter on POD 2 was significantly correlated with postoperative complications.

Operative fluid overload and inadequate pain control can be determinants of postoperative low compliance to enhanced recovery protocol [20,21,22]; however, low compliance can also be considered an early sign for underlying complications. In a series of colon cancer patients, the failure to remove the urinary catheter and to stop intravenous fluids on POD 2 was a predictive indicator of morbidity [8]. To detect an early low compliance might yield to identify patients with higher risk to develop complications afterwards. These patients could benefit from proper diagnostics and the early treatment of complications. This is very important, especially in patients with advanced age and multiple comorbidities.

Previous studies found that minimally invasive colorectal surgery had an independent role to favor early postoperative recovery, to reduce overall morbidity, and to shorten the hospital stay [9,10,23,24]. In the present series, successful laparoscopic surgery was widely performed and conversion to laparotomy was necessary in only 2.7% of patients. A multivariate analysis showed that open surgery was the most important variable associated with low compliance to enhanced recovery protocol on POD 2. Our data also suggest that the elderly and patients who were given a long surgical procedure or diverting stoma had a lower compliance rate. Therefore, a tailored approach with a tight postoperative monitoring should be performed in these patients.

The rate of low compliance on POD 2 was similar to that reported following colonic surgery [8]. The lowest protocol adherence was found for removal of the urinary catheter and the stopping of intravenous fluids, whereas the highest adherence was found for naso-gastric tube removal and oral feeding recovery. Early low compliance to postoperative protocol was significantly associated with overall morbidity and major complications. In particular, failure to remove the urinary catheter on POD 2 played an independent role to favor postoperative morbidity. A delayed removal of urinary catheter increases urinary tract infections, and reduces the patient’s mobilization favoring respiratory complications [25].

The impact of the three additional operative items was positive. Both TAP block and IMA sparing were successfully performed in all patients, allowing good pain control, no anastomotic leak, and a lymph-node collection comparable to the overall series. A failure of low pneumoperitoneum was recorded in 14% of patients. These promising results and previous reports [20,21,22] should encourage the incorporation of a TAP block, low pneumoperitoneum, and IMA sparing in the enhanced recovery protocols.

A possible limitation of the present study is that participating hospitals could differ in the degree of enhanced recovery pathway implementation. However, the high adherence to preoperative and operative items indicates that the vast majority of patients followed a comprehensive protocol. The wide range of patients’ age and ASA score suggests a small likelihood of selection bias.

## 5. Conclusions

In conclusion, early low compliance to postoperative enhanced recovery protocols was associated with overall morbidity and major complications following rectal surgery. Variables associated with early low compliance were advanced age, long procedure, open surgery and diverting stoma, suggesting a tailored and careful approach in these patients.

## Figures and Tables

**Figure 1 cancers-14-05736-f001:**
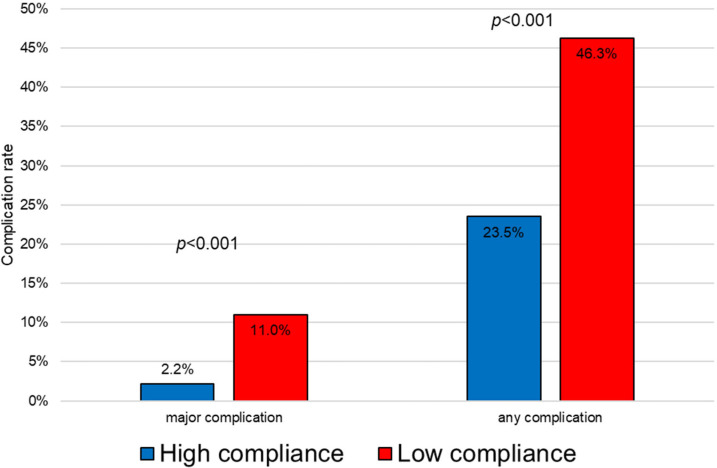
Correlation between morbidity and compliance on postoperative day 2.

**Table 1 cancers-14-05736-t001:** Patients’ characteristics.

Variable		Median	IQR	*N*	%
Age		68.00	59.76–76.5		
Sex	M			276	62.9%
	F			163	37.1%
BMI		24.82	22.59–27.68		
BMI class	<25			223	50.8
	25–29			164	37.4
	>30			52	11.8
ASA score	1			60	13.7%
	2			238	54.2%
	3			127	28.9%
	4			14	3.2%
Diabetes				53	12.1%
Preoperative Haemoglobin		13.40	12.2–14.5		
Neoadjuvant	CT-RT			113	25.7%
Mechanical bowel preparation				172	39.3%
Surgery	Anterior resection			403	91.8%
	Abdominoperineal amputation			36	8.2%
Duration of Surgery (min)		243	191–300		
Intraoperative inotropes				23	5.3%
Successful laparoscopy				363	82.7%
Laparoscopy converted to open surgery				10	2.7%
Diverting Stoma				241	54.9%
Drain				345	78.8%

**Table 2 cancers-14-05736-t002:** Cumulative compliance with postoperative era items.

Item	POD				
	0	1	2	3	4
Naso-gastric tube removal	90.8	96.6	97.5	99.1	99.5
Solid Diet	7.3	60.0	81.2	91.8	95.4
Stop IV infusion	1.6	41.2	65.0	79.2	85.4
Urinary Catheter removal	1.7	41.1	64.5	84.9	91.7
Mobilization > 4 h	8.0	43.2	65.0	74.6	81.4

**Table 3 cancers-14-05736-t003:** Variables associated with low compliance on postoperative day 2.

Variable		Postoperative Compliance				Univariate Analysis				Multivariate Analysis			
		High Compliance		Low Compliance		OR	95% CI		Sign.	OR	95% CI		Sign.
		N/Median	%/IQR	N/Median	%/IQR								
Men		67	38.1%	36	40.9%	1.126	0.668	1.899	0.656				
Age (years)		67	58.22–75	71.95	61–77.74	1.032	1.007	1.057	0.011	1.036	1.006	1.067	0.018
BMI		24.82	22.2–27.39	24.10	22.12–27.14	0.992	0.926	1.062	0.808				
BMI class	<25	89	51.4%	51	58.0%	1 (ref)							
	25–29	65	37.6%	27	30.7%	0.725	0.412	1.276	0.265				
	>30	19	11.0%	10	11.4%	0.918	0.397	2.127	0.843				
ASA score	1	27	15.3%	10	11.4%	1 (ref)							
	2	99	56.3%	48	54.5%	1.309	0.586	2.923	0.511				
	3	42	23.9%	29	33.0%	1.864	0.784	4.433	0.159				
	4	8	4.5%	1	1.1%	0.338	0.037	3.052	0.334				
Diabetes		15	8.5%	14	15.9%	2.031	0.932	4.424	0.075				
Haemoglobin (g/dL)		13.70	12.8–14.6	13.10	12.3–14.3	0.829	0.696	0.989	0.037	0.913	0.742	1.124	0.392
Neoadjuvant CT/RT		36	20.5%	19	21.6%	1.071	0.573	2.003	0.830				
Mechanical bowel preparation		71	40.6%	33	37.5%	0.879	0.519	1.488	0.631				
Preoperative glucidic drink		108	61.4%	59	67.0%	1.281	0.748	2.194	0.367				
Epidural catheter		47	26.9%	26	29.5%	1.142	0.648	2.013	0.646				
Intraoperative advanced volemia monitoring		59	33.7%	13	14.8%	0.341	0.175	0.664	0.002	0.48	0.222	1.036	0.062
Operative inotropes		5	2.9%	4	4.5%	1.619	0.424	6.186	0.481				
Operative warming		172	98.3%	88	100.0%	1.760	0.769	2.356	0.897				
Duration of Surgery		215	175.5–275	263	210–317.5	1.006	1.003	1.009	0.000	1.006	1.002	1.010	0.002
Open Surgery		15	8.5%	20	22.7%	3.157	1.526	6.531	0.002	2.732	1.173	6.362	0.02
Abdominoperineal amputation		12	6.8%	12	13.6%	1 (ref)							
Anterior resection		164	93.2%	76	86.4%	0.463	0.199	1.079	0.074				
Diverting stoma		70	39.8%	58	65.9%	2.928	1.716	4.995	0.000	1.907	1.033	3.518	0.039
Drain		141	80.1%	76	86.4%	1.572	0.771	3.206	0.213	1.293	0.573	2.916	0.536

ref: reference.

**Table 4 cancers-14-05736-t004:** Patient’s outcomes.

Variable		Median	IQR	*N*	%
Postoperative Pain (NRS)	POD 1	2	1–4		
	POD 2	2	0–3		
	POD 3	1	0–2		
	POD 4	0	0–1		
Overall morbidity				149	32.6%
Major complication				27	6.2%
Clavien-Dindo grade	0			290	67.0%
	1			54	12.5%
	2			44	10.2%
	IIIa			18	4.2%
	IIIb			22	5.1%
	Iva			3	0.7%
	Ivb			2	0.5%
	V			0	0.0%
Anastomotic leak				27	6.2%
Abdominal abscess				8	1.8%
Respiratory complication				11	2.5%
Wound infection				16	3.7%
Urinary infection				12	2.8%
Reoperation				24	5.5%
Readmission				13	3.0%
Day Fit for Discharge		5	4–8		
Length of stay		6	5–8		

**Table 5 cancers-14-05736-t005:** Variables associated with any complication.

Variables		Univariate Analysis				Multivariate Analysis			
		OR	95% CI		Sign.	OR	95% CI		Sign.
ASA score 3–4		1.211	0.793	1.849	0.375				
Age		1.000	0.983	1.018	0.968				
Men		1.513	0.990	2.312	0.056				
Diabetes		1.420	0.787	2.564	0.245				
BMI < 25		1 (ref)							
BMI 25–29		1.142	0.735	1.774	0.555				
BMI ≥ 30		1.707	0.918	3.175	0.091				
Neadjuvant CT/RT		1.388	0.888	2.170	0.150				
Mechanical bowel preparation		1.037	0.690	1.561	0.860				
Surgery	Anterior resection	1 (ref)							
	Abdominoperineal amputation	1.107	0.529	2.318	0.787				
Successful_laparoscopy		0.693	0.416	1.155	0.159				
Failure to remove NG tube on POD2		1.784	0.535	5.949	0.346				
Failure to have solid diet on POD2		2.113	1.293	3.452	0.003	1.357	0.737	2.498	0.327
Failure to stop IV fluids on POD2		2.191	1.445	3.321	0.000	1.518	0.895	2.574	0.122
Failure to remove urinary catehter on POD2		2.359	1.550	3.591	0.000	1.806	1.133	2.878	0.013
Failure to mobilize >4 h on POD 2		1.835	1.206	2.793	0.005	1.466	0.936	2.295	0.095
Poorly controlled pain on POD2 (NRS > 3)		1.882	1.142	3.101	0.013	1.430	0.839	2.438	0.189

ref: reference.

**Table 6 cancers-14-05736-t006:** Monza subgroup patients’ characteristics.

Variable	Value	Median	IQR	*N*	%
Men				18	51
Age		71	61.5–80		
BMI class	<25			16	46
	25–29			11	31
	≥30			8	23
ASA score	1			2	5.7
	2			13	37
	3			19	54
	4			2	5.7
IMA sparing				35	100
TAP-block				35	100
Failure Low Pneumop.				5	8.6
Successful laparoscopy				35	100
Lymph nodes harvested		15	12–21		
NRS	POD 1	2	2–5		
	POD 2	2	2–5		
	POD 3	1	1–3		
	POD 4	0	0–2		
Mobilization > 4 h POD2				32	91
Clavien Dindo	0			21	60
	1			4	11
	2			6	17
	3a			1	2.8
	3b			1	2.8
	4			0	0
	5			0	0
Anastomotic leak				0	0
Fit for discharge, d		6	4–7		
LOS, d		7	5–8		
Readmission				3	8.6

## Data Availability

Data can be obtained under request to the corresponding author.

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
