# Peer review of "Early Postoperative Low Compliance to Enhanced Recovery Pathway in Rectal Cancer Patients"

_cancers, 2022, doi:10.3390/cancers14235736_

Round 1

Reviewer 1 Report

I read the article :" early postoperative low compliance to enhanced recovery pathway in rectal cancer"  with interest. the authors are to be congratulated for the quality of this study which reports interesting results for rectal surgeons. the series is important and the discussion is interesting emphasizing that poor compliance can be an early sign of complication

Author Response

Thank you for your comments

Reviewer 2 Report

The author conducted a prospective observational study comparing the relationship of patients’ compliance to ERAS protocol and surgical outcomes in those with rectal cancer underwent surgery. This looks to be a good study and interesting outcomes. However, there are something need to be addressed. 

1. This is a study regarding rectal cancer patients, however, the author only mentioned 25.7% of patients underwent neoadjuvant treatment in the study cohort. There was no mention of distribution of rectal cancer staging and distance from anal verge. This could be the confounder affect the patient’s outcome and compliance and should be addressed.

2. It seems like table 2 and table 4 are misplaced. 

3. Is there any combined surgery in the study cohort? Which may influence the surgery time and patient’s recovery and thus should be mentioned. 

4. In Table 6: MA sparing should be IMA sparing.

5. The definition of length of stay should be defined in the methods section. In the study, the median length of stay is 6 days in the whole cohort and it looks a little longer than previous studies reporting ERAS in CRC patients. The author should discuss this more as the study only include rectal cancer patients and the difference of this study compared to others should be addressed. 

Author Response

The author conducted a prospective observational study comparing the relationship of patients’ compliance to ERAS protocol and surgical outcomes in those with rectal cancer underwent surgery. This looks to be a good study and interesting outcomes. However, there are something need to be addressed. 

1. This is a study regarding rectal cancer patients, however, the author only mentioned 25.7% of patients underwent neoadjuvant treatment in the study cohort. There was no mention of distribution of rectal cancer staging and distance from anal verge. This could be the confounder affect the patient’s outcome and compliance and should be addressed.

Answer: Unfortnately our register was not designed for oncological purposes and does not provide these data. All patients underwent anterior rectal resection or abdominoperineal amputation. We think that the distance of the cancer from the anal verge has no effect on the postoperative course but only in the choice of intervention (anterior resection vs APR) and therefore we decided not to retrieve also these data.

2. It seems like table 2 and table 4 are misplaced. 

Thank you for your comment; tables are misplaced and the correct number has been fixed.

3. Is there any combined surgery in the study cohort? Which may influence the surgery time and patient’s recovery and thus should be mentioned. 

The study did not include combined resections in order to reduce confounders. The method section was updated.

4. In Table 6: MA sparing should be IMA sparing.

Thank you, table was corrected.

5. The definition of length of stay should be defined in the methods section. In the study, the median length of stay is 6 days in the whole cohort and it looks a little longer than previous studies reporting ERAS in CRC patients. The author should discuss this more as the study only include rectal cancer patients and the difference of this study compared to others should be addressed. 

Length of stay was defined as the number of days in hospital from the intervention to the discharge at home. The length of stay could be affected by several factors as the older age, comorbidities, frailty and organizationa needs. For this reason the most importat indicator that should be conisdere is the Time to readiness to discharge (TRD), indicated in results and tables that is 5 days, similar to data reported in literature. TRD is described in the method section. For this reason we think is not important to discuss the longer length of stay, adding confusion to readers.

Round 2

Reviewer 2 Report

The correction is fine. Congratulations for the study.